# The Effect of COVID-19 Perceived Risk on Internet Addiction among College Students in China: An Empirical Study Based on the Structural Equation Model

**DOI:** 10.3390/ijerph192013377

**Published:** 2022-10-17

**Authors:** Ling Pan, Jun Li, Ziao Hu, Henan Wu

**Affiliations:** 1School of Finance and Economics, Hainan Vocational University of Science and Technology, Haikou 571126, China; 2Department of Education Management, Chinese International College, Dhurakij Pundit University, Bangkok 10210, Thailand

**Keywords:** Chinese college students, COVID-19 perceived risk, Internet addiction, difficulties in emotion regulation

## Abstract

This study focused on COVID-19 perceived risk and Internet addiction among Chinese college students during the lockdown. On the basis of the Social Cognitive Theory, this study proposed a mediating model to evaluate the mediating role of difficulties in regulating emotion between the COVID-19 perceived risk and Internet addiction. A questionnaire survey was conducted among 690 college students during the COVID-19 lockdown in China. The results showed that the COVID-19 perceived risk was significantly positively associated with Internet addiction (r = 0.236, *p* < 0.001) and difficulties in emotion regulation (r = 0.220, *p* < 0.001), difficulties in emotion regulation was significantly positively associated with Internet addiction (r = 0.368, *p* < 0.001). The COVID-19 perceived risk had a significant and positive predictive effect on Internet addiction (β = 0.233, *p* < 0.001) among Chinese college students. The analysis of the mediation model showed that difficulties in emotion regulation partially mediated the relationship between COVID-19 perceived risk and Internet addiction (indirect effect value was 0.051 with 95% Confidence Interval ranging from 0.027 to 0.085). The findings not only enhanced our understanding of the internal influence mechanism of COVID-19 perceived risk on Internet addiction but also provided a practical basis for college education works. Finally, discussions and suggestions were provided on the basis of the results.

## 1. Introduction

Internet addiction is defined as symptoms of compulsive or excessive Internet use [1,2], also known as “problematic Internet use” [3]. Particularly, it refers to the inability to control the use of the Internet [4], characterized by excessive or uncontrollable computer use and Internet access which can produce abstract possessiveness, impulses, or behaviors causing damage or distress [5]. Internet addiction mainly includes excessive participation in activities such as playing online games, browsing social networking sites, compulsive online shopping and watching and downloading irrelevant videos [6]. Previous studies have confirmed that Internet addiction can cause serious harm to individuals in several aspects, such as inability to express emotions [7], depression [8,9], anxiety [9,10], sleep disorders [9,10,11], social dysfunction [9], and even suicide [12]. Therefore, Internet addiction has attracted considerable attention from researchers and the public worldwide [13]. In addition, because of psychological and developmental characteristics, college students are particularly prone to Internet addiction [14]. Chinese college students are more prone to Internet addiction than students from other countries [15,16], which is extremely alarming [16]. Data from a 2016 study of 1173 Chinese college students showed that up to 15.2 percent of college students suffer from Internet addiction [17]. With the outbreak of COVID-19, Internet addiction among Chinese college students has further intensified [18,19], and the rate of Internet addiction has increased to 23.7% [18]. Therefore, it is particularly critical to identify the impact factors of Internet addiction among Chinese university students during the COVID-19 pandemic.

At present, the COVID-19 epidemic may still cause great bodily harm or even death [20]. In order to contain the spread of the epidemic, the Chinese government has implemented extremely strict prevention and control policies, such as: Closed all schools [21], confined students to their homes [22] or isolated college students from the outside world by confining them to their schools [23]. The virus was highly transmissible and contagious [24], causing “fear” [25] and negative emotions among Chinese college students [26]. Previous studies have suggested that individuals’ perception of risk is one of the drivers of Internet addiction behavior [27], and it may also cause individuals’ difficulties in emotion regulation [28]. Recent studies have confirmed that difficulties in emotion regulation are also an important contributing factor to Internet addiction among college students [29,30,31].

This study selected college students under lockdown in China, which was highly targeted. According to the social cognitive theory, the COVID-19 perceived risk was regarded as an environmental factor, difficulties in emotion regulation as an individual factor, and Internet addiction as a behavioral factor. Comprehensively explore how the COVID-19 perceived risk affects their Internet addiction behavior. Finally, according to the results of this study, specific suggestions were provided for colleges and universities on how to deal with emergency public health events similar to COVID-19, and how to prevent and improve college students’ Internet addiction behaviors during the COVID-19 epidemic.

### 1.1. Social Cognitive Theory

The Social Cognitive Theory (SCT) proposed by Bandura (1986) [32] holds that environmental, individual, and behavioral factors influence each other. SCT has been widely used in related studies on Internet addiction [33,34,35]. Specifically, Lin et al. (2008) [33] used SCT to study the influence of the outcome expectancy and refusal self-efficacy of Internet use on Internet addiction among 4456 college students. Their study results showed that the positive outcome expectancy of Internet use positively predicted Internet addiction through refusal self-efficacy of Internet use. Moreover, it was surprising that the negative outcome expectancy of Internet use had a positive predictive effect on Internet addiction directly and indirectly through the refusal self-efficacy of Internet use.

In addition, in another study based on SCT, Lin et al. (2018) [36] constructed a mediation model of social influence (environmental factors) affecting Internet addiction severity (behavior factors) through positive outcome expectations (individual factors) among 1922 high school students in Taiwan. Moreover, Wu et al. (2013) [34] conducted a study based on SCT, indicating that people who spend more time on social networks have a higher tendency to become addicted. Addiction tendency was positively associated with outcome expectation and impulsivity but negatively with Internet self-efficacy. On the basis of SCT, a recent study conducted by Yang (2020) [35] investigated the influence of self-efficacy and self-control (personal cognitive factors) on Internet addiction in adolescents (behavioral factor) through social support (environmental factor).

Therefore, based on the SCT, this study considered COVID-19 perceived risk an environmental factor, difficulties in emotion regulation a personal factor, and Internet addiction a behavioral factor to construct a mediation model to study the effect of the COVID-19 perceived risk on Internet addiction among Chinese college students through difficulties in emotion regulation.

### 1.2. COVID-19 Perceived Risk and Internet Addiction

Perceived risk refers to an individual’s subjective perception and self-awareness of various objective risks in the external environment and emphasizes the influence of an individual’s experience gained from subjective perception and direct judgment on cognition, thus forming a subjective judgment and assessment of the possibility, controllability, and severity of consequences and other attributes of risk occurrence [37,38]. It is crucial in determining individual behavior in response to global pandemics such as severe acute respiratory syndrome [39]. At present, COVID-19, also recognized as a global pandemic, has caused a widespread risk perception among people in various countries [40]. In a study conducted in Turkey, participants reported to generally feel the risk of the epidemic during the COVID-19 outbreak, with women feeling more at risk than men [40]. In another study of 1379 respondents in Indonesia, unmarried people were found to have a higher risk perception than married people, and young people had the highest perception of COVID-19 risk [41]. Meanwhile, a recent study found that college students are the group most affected by the COVID-19 pandemic [42]. Therefore, the risk perception of COVID-19 among college students has attracted considerable attention from researchers [42,43,44].

In addition, studies have confirmed that risk perception motivates coping behaviors [40]. COVID-19, an epidemic, spreads extremely fast and is highly contagious [24]. People can commonly perceive the risk of contracting COVID-19 [45] and adopt several risk coping behaviors, such as maintaining social distance, closing schools (lockdown), staying at home, and restricting movement to reduce its spread [6,40,46]. Because of self-isolation and widespread lockdown, people had to shift most of their work and life activities online [4], leading to heavy Internet use and possibly Internet addiction [4]. Survey data revealed that different groups of individuals, including college students, showed varying degrees of Internet addiction during the epidemic [47,48]. One of these studies, conducted in Mexico, found that 62.7% of respondents experienced different degrees of Internet addiction a few weeks after the COVID-19 lockdown [47]. A survey from China reported that the total time spent on the Internet by vulnerable groups, such as college students, significantly increased during the COVID-19 pandemic compared with previous periods and increased Internet addiction [48]. Another study in Nigeria noted that college students were at an increased risk of Internet addiction during the COVID-19 epidemic and found higher levels of Internet addiction among students in EKSU (public universities) or liberal arts colleges [46]. Several recent studies have shown that the COVID-19 perceived risk is positively associated with Internet addiction in different populations [4,23,27,48]. One study conducted in outbreak-controlled regions in China further revealed that the COVID-19 risk perception was significantly positively associated with Internet addiction among college students experiencing campus lockdown, and COVID-19 perceived risk could positively predict Internet addictive behavior [23].

On the basis of the above discussion, this study speculated that COVID-19 perceived risk may be a key influencing factor of Internet addiction among Chinese college students during the lockdown. Therefore, hypothesis 1 (H1) was put forward: COVID-19 perceived risk is significantly positively associated with Internet addiction, and it significantly positively predicts Internet addiction among college students.

### 1.3. Mediating Role of Difficulties in Emotion Regulation

Difficulties in emotion regulation are also known as emotion dysregulation [49,50]. Emotion regulation is a process in which individuals manage and modify their own or others’ emotions, including awareness and understanding of emotions, acceptance of emotional experience, control of emotional expression behaviors, and flexible selection of appropriate emotional regulation strategies [51]. Its function is to select acceptable responses in the social context to adjust incompatible conflicts and achieve self-internal balance [52]. Once the emotional regulation ability of individuals is impaired, they will have difficulties with emotional regulation [53,54], leading to social adaptation problems and internal self-imbalance [55].

A previous study has indicated that the perceived risk of pandemics (such as Ebola) is associated with higher negative emotions [56]. Recent studies have confirmed that COVID-19, which like Ebola, is defined as a pandemic by the World Health Organization, can cause individuals to experience negative emotions when they perceive a severe threat of COVID-19 during this period [26,57,58]. Stanley and Larsen (2021) [59] indicated that when a person’s negative emotion is quite large, it may go beyond the range of human adaptation, resulting in emotion dysregulation or difficulty in emotion regulation. A study confirmed that as soldiers often operate in combat environments, they are often faced with the threat of death, generating negative emotions of insecurity and vulnerability in soldiers, thus continuously depleting their emotional regulation ability, which eventually may lead to emotion dysregulation [59]. Another study of 846 volunteers in Israel found that because of changes in daily life and general concern about the risk of infection and even death, participants may find themselves unable to effectively control and regulate their emotions when facing the great risk of COVID-19 [28].

A previous study has linked emotional states with addictive behaviors [60]. Several studies have indicated that difficulties in emotion regulation are a potential common influencing factor of addictive behaviors, such as chemical addictions, including smoking, alcoholism, or drug addiction [61,62], risky sexual behavior [63], gambling addiction [64], video game addiction [65], social network addiction [66], and Internet addiction [67]. A study has shown that people with insufficient emotion regulation ability are more inclined to take activities that can bring immediate pleasure [68] to relieve their pain [69]. Under certain circumstances, individuals may overuse the Internet to avoid negative effects [70] and as a coping strategy to compensate for emotional regulation defects [71]. The outcomes may range from feeling fun to losing control and leading to obsession [72]. Therefore, people with lower emotional regulation abilities are more likely to have problems using the Internet [73]. The study results of Ye et al. (2017) [74] supported the above findings and found that college students with difficulties in emotion regulation would satisfy their needs through mobile phone networks for a long time. In addition, several studies have indicated that difficulties in emotion regulation are significantly associated with Internet addiction [65,75,76,77]. Another study clearly indicated that difficulties in emotion regulation would make it difficult for college students to control their use of the Internet, leading to Internet addiction [67]. Multiple studies conducted on college students during COVID-19 have also confirmed that emotional dysregulation or difficulties in emotion regulation are a predictor of Internet addiction [29,30,31].

Furthermore, studies during the pandemic have often discussed difficulties in emotion regulation or emotional dysregulation as a mediator for predicting addictive behaviors [31,78,79,80]. For example, the study by Hegbe et al. (2021) [79] showed that emotional dysregulation played a mediating role in the relationship between depression to sex addiction. Another study of 231 students discovered that anxiety sensitivity could indirectly predict the tendency to addiction through difficulties in emotion regulation [80]. Lim et al. (2020) [78] considered childhood unhappiness an environmental factor, social network addiction a behavioral factor, and difficulties in emotion regulation a personal factor in their study. The results indicated that difficulties in emotion regulation play a crucial mediating role in the effect of childhood unhappiness on social network addiction. A Turkish study found that social anxiety could affect excessive internet use among college students through difficulties in emotion regulation [31].

Therefore, this study speculated that the COVID-19 perceived risk might significantly positively affect the Internet addiction of college students through difficulties in regulating emotion. Hypothesis (2) was proposed: COVID-19 perceived risk is significantly positively associated with difficulties in emotion regulation, difficulties in emotion regulation is significantly positively associated with Internet addiction and has a mediating effect on the relationship between COVID-19 perceived risk and Internet addiction in college students.

On the basis of the above theoretical analysis and literature discussion, this study constructed a hypothesis model, which is shown in Figure 1:

## 2. Materials, Procedure and Methods

### 2.1. Participants

We adopted convenience sampling at a university located in south China. The sampling university is representative due to its location at the border of China, where the epidemic is more unstable and challenging to control. Therefore, local universities often apply different levels of containment policies to prevent the risk of epidemic spreading. Students, who were interested in the topic of the present study at the sampling university, were recruited voluntarily. Data were collected, submitted, and analyzed anonymously after obtaining informed consent from the participants, following the principles of the Declaration of Helsinki [81].

### 2.2. The Procedure of Data Collection

The Academic Ethics Committee of Hainan Vocational University of Science and Technology approved the present study (HKD-2022-25). Data were collected from 10–13 May 2022, when the university was under a state of lockdown, and the activities of university students were restricted. As recommended by Podsakoff et al. (2003) [82], the teachers in charge of questionnaire distribution were professionally trained before data collection to prevent information bias, by which they were required to inform the participants about the purpose of the study and explain the information about the questionnaire. Participants could refuse or withdraw if they had doubts during the process. Questionnaires were distributed through Questionnaire Star (www.wjx.cn (accessed on 13 May 2022)). A total of 750 questionnaires were distributed and the average time was 8 min. Questionnaires that completion times were too short (<3 min) and too long (>15 min), or incomplete, were excluded.

After data cleaning and screening, 690 valid questionnaires remained, with a return rate of 92%. According to the criteria of sample size proposed by Israel (1992) [83], *n* = 30,000 for the sampling unit of this study, so the sample size should be no less than 652, and the adequate sample size of 690 in this study exceeded the above criteria. Among them, 309 (44.8%) were first-year university students, 160 (23.2%) were second-year university students, 148 (21.4%) were third-year university students, 30 (4.3%) were fourth-year university students, and 43 (6.2%) were graduate students; 170 (24.6%) male students and 520 (75.4%) female students; 229 (33.2%) were only-child status and 461 (66.8%) were non-only-child status; 166 (24.1%) were ethnic minorities and 524 (75.9%) were China’s main nationality. The ages of all participants were between 18 and 26 years.

### 2.3. Measures

#### 2.3.1. COVID-19 Perceived Risk

The public risk perception scale for public health emergencies developed by Dai et al. (2020) was used to assess participants’ perceived risk of the COVID-19 pandemic. The scale was tested to have good credibility in a Chinese sample [38]. The scale consists of four dimensions: severity of the epidemic (e.g., I think the epidemic is severe in my region), controllability (e.g., I think the epidemic and its spread are difficult to control), the severity of the health effects (e.g., once infected, I think the impact on the body is extremely severe), and possibility (e.g., I have a great possibility of being infected), a total of 10 questions. A 5-point Likert scale was used, with scores ranging from 1 (totally agree) to 5 (totally disagree). Higher scores indicated a higher level of COVID-19 perceived risk.

#### 2.3.2. Internet Addiction

A 4-item scale to assess Internet addiction during COVID-19 [4] was used in the present study, which had good reliability and validity in the prior study. It was a single-dimension scale (e.g., During the Pandemic, I always wanted to use the Internet). All Responses were rated on a 7-point Likert scale with scores ranging from 1 (strongly disagree) to 7 (strongly agree). Higher scores indicated more severe Internet addiction.

#### 2.3.3. Difficulties in Emotion Regulation

We use the brief version of the difficulties in emotion regulation scale (DERS-16) to assess the participants’ difficulties in regulating emotions [84]. It was a 16-item scale with five dimensions: lack of emotional clarity (e.g., I am confused about how I feel); difficulties engaging in goal-directed behavior (e.g., When I am upset, I have difficulty getting work done); impulse control difficulties (e.g., When I am upset, I become out of control); limited access to effective emotion regulation strategies (e.g., When I am upset, I believe that I will remain that way for a long time.); nonacceptance of emotional responses (e.g., When I am upset, I feel ashamed with myself for feeling that way.). All responses were rated on a 5-point Likert scale with scores ranging from 1 (almost never) to 5 (almost always)—higher scores indicating more difficulty with regular emotion.

### 2.4. Statistical Analyses

The statistical strategy for this study consisted of two parts. First, SPSS 21.0 was used for descriptive statistics of the sample, descriptive statistics and correlation analysis of the variables, testing the reliability of the measurement instruments, and assessing the common method variance (CMV). Second, AMOS 21.0 was used to assess the measurement model’s validity and test the hypothesis model. Specifically, as Anderson and Gerbing (1988) [85] suggested, we first tested the structural validity of the measurement models for the three variables through Confirmatory Factor Analysis (CFA). Then the fit and path coefficients of the hypothesized mediation model were assessed by the Structural Equation Model (SEM). Finally, for the rigor of the results, the bias-corrected nonparametric percentile Bootstrap method was used to verify the confidence intervals (CI) of the indirect, direct, and total effects, where the exclusion of 0 showed the significance of the partial mediating effect [86].

### 2.5. Common Method Variance (CMV) Test

The CMV was assessed by Harman’s one-factor test. Unrotated factor analysis indicated that the Kaiser-Meyer-Olkin (KMO) was 0.928 (>0.8), and the Bartlett test of sphericity reached significance (*p* < 0.001). A total of 4 factors were extracted from the factor analysis, and the explanatory power of the first factor was 38.001%, which did not exceed 50% [82], indicating that the CMV problem did not affect the study results.

## 3. Results

### 3.1. Reliability and Validity Assessment of Measurement Model

#### 3.1.1. COVID-19 Perceived Risk

The CFA results indicated that the second item of the first dimension of the scale had factor loadings less than 0.5 and was removed [87]. The CFA results was shown in Table 1. After deleting this item, the factor loadings were between 0.542 and 0.819(>0.5), the composite reliability (CR) values were between 0.603 and 0.796(>0.6), and the average variance extracted (AVE) values were between 0.439 and 0.646(>0.5). According to Fornell and Laecker (1981), even if the AVE is less than 0.5, the convergent validity of the scale is still acceptable under the conditions of the CR value meeting the criteria (greater than 0.6). All the values met the standard value [88]. Therefore, the convergent validity of the COVID-19 perceived risk scale was appropriate. The Cronbach’s α of each dimension was between 0.600 and 0.795, greater than 0.6, indicating the excellent reliability [89].

#### 3.1.2. Internet Addiction

As shown in Table 2, the factor loadings were between 0.620 and 0.946(>0.5), the CR value was 0.873(>0.7), and the AVE value was 0.639(>0.5). All the values exceeded the standard value and indicated the high convergent validity of the Internet addiction scale [88,90]. The Cronbach’s α was 0.871, greater than 0.7, indicating the excellent reliability [91].

#### 3.1.3. Difficulties in Emotion Regulation

As shown in Table 3, the factor loadings were between 0.778 and 0.949(>0.5), the CR values were between 0.879 and 0.958(>0.7), and AVE values were between 0.708 and 0.884(>0.5). All the values exceeded the standard value and indicated the high convergent validity of the DERS-16 [88,90]. The Cronbach’s α of each dimension was between 0.876 and 0.958, greater than 0.7, indicating the excellent reliability [91].

### 3.2. Discriminant Validity

As shown in Table 4, the square root of AVE of each dimension was greater than the correlation coefficient of each dimension which indicated the high discriminant validity [88].

### 3.3. Descriptive Statistics and Correlation Analysis

Descriptive statistics and correlation analysis were shown in Table 5. The statistical results showed that college students under lockdown had a high level of COVID-19 perceived risk (mean = 3.306), Internet addiction (mean = 4.694), and difficulties in emotion regulation (mean = 2.534) in the present study.

In addition, the results indicated that the COVID-19 perceived risk and internet addiction had significant positive correlation (r = 0.236, *p* < 0.001); the COVID-19 perceived risk and difficulties in emotion regulation had significant positive correlation (r = 0.220, *p* < 0.001); difficulties in emotion regulation and Internet addiction had significant positive correlation (r = 0.368, *p* < 0.001).

### 3.4. Structural Model

The second-order SEM was used to construct the mediation model and to test the mediation effect of difficulties in emotion regulation. The fit indicators of the hypothetical mediation model were: CMIN/DF = 3.150, RMSEA = 0.056, GFI = 0.885, CFI = 0.953, NFI = 0.933, TLI = 0.948, IFI = 0.953, and PNFI = 0.838, indicating that the fit of the structural equation model was better [92].

The results of the study were shown in Figure 2, where COVID-19 perceived risk significantly and positively predicted Internet addiction (β = 0.233, *p* < 0.001); COVID-19 perceived risk also significantly and positively predicted difficulties in emotion regulation (β = 0.216, *p* < 0.001); difficulties in emotion regulation significantly and positively predicted Internet addiction (β = 0.237, *p* < 0.001). The result indicated that difficulties in emotion regulation partially mediated the relationship between the COVID-19 perceived risk and internet addiction.

The bias-corrected nonparametric percentile bootstrap method further examined the mediating effect of difficulties in emotion regulation. The results were shown in Table 6. The indirect effect value was 0.051 with 95% CI ranging from 0.027 to 0.085, without 0, indicating a significant mediating effect. The direct effect value was 0.233 with 95% CI ranging from 0.136 to 0.329, excluding 0, indicating a partial mediating effect. The total effect value was 0.284 with 95% CI ranging from 0.186 to 0.380, excluding 0. The mediation effect accounted for 17.96% of the total effect.

## 4. Discussion

The results verified hypothesis 1, i.e., COVID-19 perceived risk is significantly positively associated with Internet addiction, and it can significantly positively predict Internet addiction in college students. This finding is similar to that of existing studies [23,48], verifying that the COVID-19 perceived risk is associated with Internet addiction [23,48] and is the precipitating factor of Internet addiction [23]. This may be because of the following: first, college students may have felt the threat to their health and life during the COVID-19 pandemic [42,43,44], encouraging college students to take corresponding countermeasures [40], such as self-isolation, avoiding public gatherings, maintaining physical and social distancing, and other related measures to avoid infection. Therefore, college students shifted more of their life and study on the Internet [4], greatly increasing the time of using the Internet, which may lead to Internet addiction [4,93]. Second, because of the lockdown, the uncertainty of COVID-19 [94] made college students learn more about the dynamics and development of COVID-19 through the Internet, making them rely on the Internet, thus leading to Internet addiction.

The results verified hypothesis 2, i.e., the COVID-19 perceived risk is significantly positively associated with difficulties in emotion regulation, difficulties in emotion regulation is significantly positively associated with Internet addiction and has a mediating effect on the relationship between COVID-19 perceived risk and Internet addiction in college students. Particularly, the COVID-19 perceived risk can not only directly and significantly affect Internet addiction among college students but also indirectly and positively affect Internet addiction through difficulties in emotion regulation. This result is consistent with that of previous studies that reconfirmed emotional dysregulation can promote addictive behaviors [78,79,80]. In a COVID-19 environment, college students generally perceive great risks [42,43,44], which may aggravate the negative emotion of college students beyond their adaptive range, leading to emotional dysregulation [59]. Under emotional dysregulation, college students may take measures that bring immediate pleasure [68] and try to escape realistic negative emotions by overusing the Interne [70] to alleviate the pain caused by negative emotion [77]; however, the outcome may range from happiness on the Internet to losing control and finally falling into obsession [72]. Moreover, this supported that difficulties in emotion regulation are the crucial mediator of addictive behaviors during the pandemic [80].

This study supported the SCT. Particularly, the mediation role of difficulties in emotion regulation has been discovered, and a complementary partial mediation model has been implemented. This study further discussed how COVID-19 perceived risk affects Internet addiction among college students during the lockdown. The results indicated that difficulties in emotion regulation and perceived COVID-19 risk could provide a more comprehensive interpretation of Internet addiction among college students during the epidemic. Difficulties in emotion regulation were a key promoting factor for Internet addiction among college students, which can complement COVID-19 perceived risk and further deepen Internet addiction.

There are some limitations of this study that can be carried out in future studies. First, this study included college students from a university in Yunnan Province, China, during the lockdown, preventing the generalization of the findings. Therefore, future studies may consider studying college students from different regional and cultural backgrounds worldwide to compare the risk perception and Internet addiction among college students during COVID-19 or other groups (such as graduate students) to increase the universality of the study results.

Second, this was a cross-sectional study, which examined the correlation and predictive relationship between variables; however, it could not obtain the causal relationship between variables. Future studies can consider longitudinal studies to further understand the causal relationship between the variables.

Third, this study adopted the quantitative research method of a questionnaire survey, which may have inferential limitations. Future studies should consider adding interviews or literature reviews so as to have a more specific and comprehensive understanding of the relationship among perceived risk, difficulties in emotion regulation, and Internet addiction during COVID-19.

Finally, this study was conducted in a university during the COVID-19 lockdown; therefore, other factors, such as family, were not considered. Future studies could consider other environmental factors, such as family and community.

## 5. Conclusions 

In conclusion, this study constructed a mediation model to explore the main influencing factors of Internet addiction among Chinese college students during the epidemic lockdown. A significant positive correlation was observed among COVID-19 perceived risk, difficulties in emotion regulation, and Internet addiction. The COVID-19 perceived risk has a significant positive predictive effect on Internet addiction and difficulties in emotion regulation of college students. Moreover, difficulties in emotion regulation played a mediating role between COVID-19 perceived risk and Internet addiction. This study supported the SCT. Particularly, COVID-19 perceived risk (environmental factor) affects difficulties in emotion regulation (personal factor) and then Internet addiction (behavioral factor) among college students. 

## 6. Suggestions

This study results provided empirical reference for universities to respond to emergency public health events such as COVID-19.

First, COVID-19 has been the most influential epidemic worldwide in recent years. Universities should regularly provide publicity and education on epidemic knowledge, scientifically interpret the current development of the epidemic, guide college students to correctly understand the severity of the epidemic and the threat to health and to keep rational thinking in public health events. One should prevent college students from causing unnecessary panic and negative emotion because of excessive perception of COVID-19 risk.

Second, the COVID-19 epidemic can increase the negative emotions of college students during the lockdown. This study suggested that colleges and universities should conduct training on emotion regulation ability to help college students correctly understand the importance of self-emotion regulation, improve their emotion regulation ability, and prevent college students from being addicted to the Internet due to emotional dysregulation.

Third, there may be different views on the epidemic reports on the Internet, which may mislead the perception of the COVID-19 epidemic among college students. It is suggested that during the lockdown, colleges and universities should guide students to distinguish network information scientifically. One should advocate proper use of the Internet and healthy Internet access and identify excessive use of the Internet as early as possible to prevent college students from being online for a long time, which leads to Internet addiction.

## Figures and Tables

**Figure 1 ijerph-19-13377-f001:**
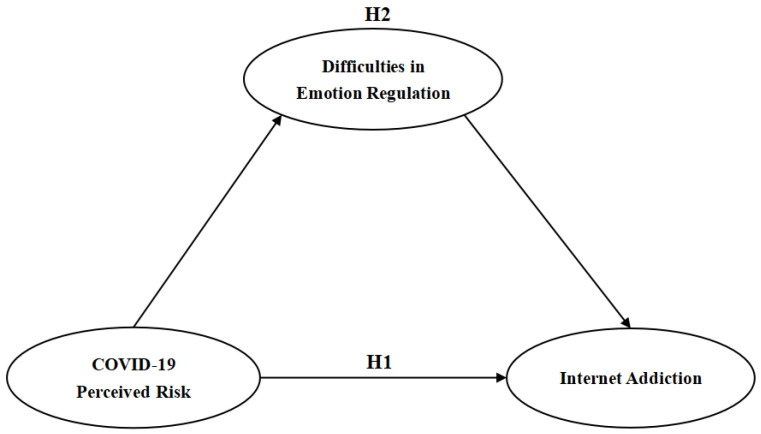
Hypothesis model.

**Figure 2 ijerph-19-13377-f002:**
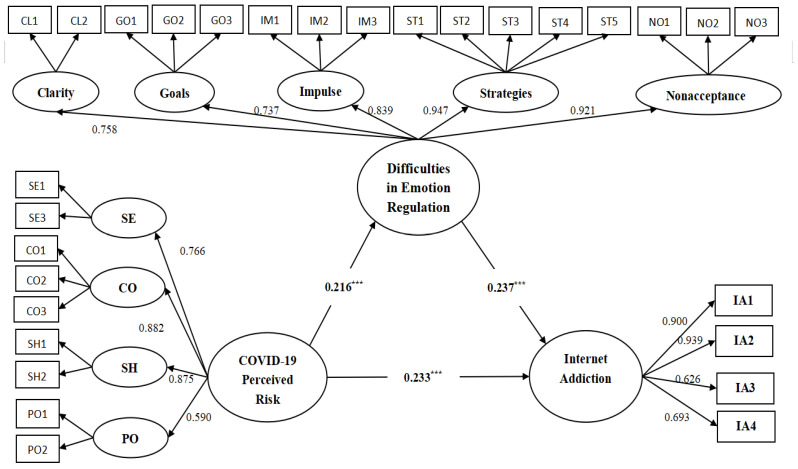
The Mediation Model. Note: *** *p* < 0.001; SE, Severity of the epidemic; CO, Controllability; SH, Severity of the health effects; PO, Possibility; IA, Internet Addiction; CL/Clarity, Lack of Emotional Clarity; GO/Goals, Difficulties Engaging in Goal-Directed Behavior; IM/Impulse, Impulse Control Difficulties; ST/Strategies, Limited Access to Effective Emotion Regulation Strategies; NO/Nonacceptance, Nonacceptance of Emotional Responses.

**Table 1 ijerph-19-13377-t001:** Reliability and validity assessment of COVID-19 perceived risk scale.

Dimension	No.	Factor Loading	CR	AVE	Cronbach’s α
Severity of the epidemic	1	0.719	0.731	0.577	0.726
	3	0.798			
Controllability	1	0.783	0.796	0.565	0.795
	2	0.762			
	3	0.708			
Severity of the health effects	1	0.788	0.785	0.646	0.785
	2	0.819			
Possibility	1	0.542	0.603	0.439	0.600
	2	0.764			

**Table 2 ijerph-19-13377-t002:** Reliability and validity assessment of internet addiction scale.

Dimension	No.	Factor Loading	CR	AVE	Cronbach’s α
Internet Addiction	1	0.898	0.873	0.639	0.871
	2	0.946			
	3	0.620			
	4	0.685			

**Table 3 ijerph-19-13377-t003:** Reliability and validity assessment of difficulties in emotion regulation scale.

Dimension	No.	Factor Loading	CR	AVE	Cronbach’s α
Clarity	1	0.850	0.891	0.803	0.888
	2	0.940			
Goals	1	0.915	0.955	0.875	0.954
	2	0.949			
	3	0.942			
Impulse	1	0.935	0.958	0.884	0.958
	2	0.943			
	3	0.943			
Strategies	1	0.899	0.954	0.807	0.954
	2	0.923			
	3	0.909			
	4	0.878			
	5	0.882			
Nonacceptance	1	0.778	0.879	0.708	0.876
	2	0.854			
	3	0.889			

Note: Clarity, Lack of Emotional Clarity; Goals, Difficulties Engaging in Goal-Directed Behavior; Impulse, Impulse Control Difficulties; Strategies, Limited Access to Effective Emotion Regulation Strategies; Nonacceptance, Nonacceptance of Emotional Responses.

**Table 4 ijerph-19-13377-t004:** Discriminant validity.

Dimension	*M*	*SD*	1	2	3	4	5	6	7	8	9	10
Severity of the epidemic	3.704	0.931	* **0.760** *									
Controllability	3.303	0.848	0.517 ***	* **0.752** *								
Severity of the health effects	3.491	0.890	0.516 ***	0.603 ***	* **0.804** *							
Possibility	2.726	0.834	0.258 ***	0.404 ***	0.351 ***	* **0.663** *						
Internet Addiction	4.694	1.282	0.200 ***	0.211 ***	0.207 ***	0.093 *	* **0.799** *					
Clarity	2.618	0.981	0.037	0.106 **	0.119 **	0.193 ***	0.256 ***	* **0.896** *				
Goals	2.933	1.033	0.174 ***	0.227 ***	0.217 ***	0.177 ***	0.450 ***	0.545 ***	* **0.935** *			
Impulse	2.306	1.058	0.031	0.141 ***	0.097 *	0.230 ***	0.246 ***	0.600 ***	0.595 ***	* **0.940** *		
Strategies	2.470	0.997	0.057	0.139 ***	0.137 ***	0.228 ***	0.328 ***	0.657 ***	0.653 ***	0.763 ***	* **0.898** *	
Nonacceptance	2.412	0.940	0.095 *	0.162 ***	0.128 **	0.243 ***	0.280 ***	0.610 ***	0.619 ***	0.707 ***	0.814 ***	* **0.841** *

Note: *n* = 690; The bold and italic numbers in the diagonal are the square root of AVE (AVE = average variance extracted); numbers in the lower diagonal denote the correlation coefficients; * *p* < 0.05, ** *p* < 0.01, *** *p* < 0.001; *M* = mean; *SD* = standard deviation.

**Table 5 ijerph-19-13377-t005:** Descriptive statistics and correlation analysis.

Variable	*M*	*SD*	COVID-19 Perceived Risk	Internet Addiction	Difficulties In Emotion Regulation
COVID-19 PERCEIVED RISK	3.306	0.674	1		
INTERNET ADDICTION	4.694	1.282	0.236 ***	1	
DIFFICULTIES IN EMOTION REGULATION	2.534	0.867	0.220 ***	0.368 ***	1

Note: *n* = 690; *** *p* < 0.001; *M* = mean; *SD* = standard deviation.

**Table 6 ijerph-19-13377-t006:** Direct effect, indirect effect, and total effect.

Path	Estimate	95% Confidence Interval	
		LL	UL
Indirect effect	0.051	0.027	0.085
Direct effect	0.233	0.136	0.329
Total effect	0.284	0.186	0.380

Note: LL: lower limit; UL: upper limit.

## Data Availability

The data presented in this study are available on request from the corresponding author.

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
