# Peer review of "The Effect of COVID-19 Perceived Risk on Internet Addiction among College Students in China: An Empirical Study Based on the Structural Equation Model"

_ijerph, 2022, doi:10.3390/ijerph192013377_

Round 1

Reviewer 1 Report

I feel happy to review this manuscript titled as "Authors have used numerical format for citation of work. I suggest authors to use APA style for citation". The topic of this article is interesting, authors have done good work. There are some changes are required. I hope it will help authors to revise their manuscript.

1. There is no need to report the measurement scale explanation in the abstract.   

2. The introduction section should be revised. I suggest authors to add research contribution, novelty of this research. There is no research objective or research question is presented in the introduction section.

3. I suggest authors to add hypotheses relationships in the Figure 1.

Good Luck

Reviewer 2 Report

Reviewer Comments

1). The numerical results or the values of the respective p-values should be included in the abstract.

2). The literature review should include the status quo before the advent of COVID-19.

3). A moderation analysis should have been better to investigate the moderating effect as shown in Figure 1.

4). The low Cronbach’s alpha of less than 0.70 obtained for Possibility is a concern.

5). Implications for research and public health could be married with the recommendations.

Reviewer 3 Report

Dear Editor and Authors,

At the beginning, I would like to thank you for my consideration as a reviewer of this manuscript. It is my pleasure to contribute to awareness of the importance of mental health.

The research explores the COVID-19 pandemic perceived risk and Internet addiction on young people. This work is well written and organized.

Introduction: There is no information about the coronavirus in China, the restrictions are not specified and there is no information about the attitudes of the citizens to this situation. I suggest to move the lines 87- 115 to the discussion section.

Methods: Please specify the inclusion and exclusion criteria of the participation in the study. Provide the age range of the participants and the socio-demographic characteristics. 

Discussion: The authors accurately discussed clinical implications and possible explanations of the results.

Limitations: this section is usually at the end of the Discussion. The authors accurately discussed the limitations and further directions of the research.
